# Exploring the Skin Brain Link: Biomarkers in the Skin with Implications for Aging Research and Alzheimer’s Disease Diagnostics

**DOI:** 10.3390/ijms241713309

**Published:** 2023-08-28

**Authors:** Stefanie Klostermeier, Annie Li, Helen X. Hou, Ula Green, Jochen K. Lennerz

**Affiliations:** 1Institute for Physiology and Pathophysiology, University of Erlangen-Nuremberg, 91054 Erlangen, Germany; stefanie.klostermeier@fau.de; 2Center for Integrated Diagnostics, Department of Pathology, Massachusetts General Hospital, Harvard Medical School, Boston, MA 02114, USA; ali32@mgh.harvard.edu (A.L.); helen.hou@hotmail.de (H.X.H.); ugreen@mgh.harvard.edu (U.G.)

**Keywords:** biomarker, diagnostics, neurodegeneration, brain stiffness

## Abstract

Neurodegenerative diseases, including Alzheimer’s disease (AD), are challenging to diagnose. Currently the field must rely on imperfect diagnostic modalities. A recent study identified differences in several key bio-mechano-physiological parameters of the skin between AD patients and healthy controls. Here, we visually align these differences with the relevant histological, aging, and embryological paradigms to raise awareness for these potential biomarkers. In a study conducted by Wu et al., a series of n = 41 patients (n = 29 with AD and n = 12 healthy controls) were evaluated, demonstrating that AD patients exhibit a less acidic skin pH, increased skin hydration, and reduced skin elasticity compared to healthy controls. We constructed a visual overview and explored the relevant paradigms. We present a visual comparison of these factors, highlighting four paradigms: (1) the findings emphasize a shared ectodermal origin of the brain and the skin; (2) functional systems such as micro-vascularization, innervation, eccrine excretory functions, and the extracellular matrix undergo distinct changes in patients with AD; (3) the human skin mirrors the alterations in brain stiffness observed in aging studies; (4) assessment of physiological features of the skin is cost-effective, accessible, and easily amenable for monitoring and integration with cognitive assessment studies. Understanding the relationship between aging skin and aging brain is an exciting frontier, holding great promise for improved diagnostics. Further prospective and larger-scale investigations are needed to solidify the brain-skin link and determine the extent to which this relationship can be leveraged for diagnostic applications.

## 1. Introduction

Diagnosing neurodegenerative diseases such as Alzheimer’s disease (AD) remains challenging. We currently rely on a series of imperfect diagnostic modalities [1,2,3]. While postmortem examination of the brain remains the gold standard, the possibility of intra-vital and importantly non-invasive diagnostic alternatives is of broad interest to the field [1,2,3]. Notably, recent work emphasized a fascinating relationship between central nervous system function and a series of bio-mechano-physiological properties of the skin in affected patients [4].

Here, we visualize this intriguing relationship (Figure 1), raise awareness for this interesting approach, and discuss translational research implications toward improved diagnostics.

## 2. Findings

In brief, the recent study by Wu et al. delineated that cutaneous bio-mechano-physiological properties differed in a series of n = 41 patients (n = 29 with AD and n = 12 healthy controls) [4]. AD was diagnosed clinically through a comprehensive assessment that included medical history, cognitive testing, functional assessment, brain imaging, and laboratory tests to rule out other potential causes of cognitive impairment. Specifically, the authors diagnosed AD following the Diagnostic and Statistical Manual of Mental Disorders (DSM)-IV criteria for dementia [5] and diagnostic criteria for probable AD, proposed by NINCDS-ADRDA (National Institute of Neurological and Communicative Disorders and Stroke–and–Alzheimer’s Disease and Related Disorders Association) [6]. The authors also employed APOE genotyping and a battery of skin tests including a multi skin test center, a laser Doppler flowmetry, and a capillary microscopy after 30 min of acclimatization at room temperature and pre-defined humidity.

Briefly, skin hydration was measured using the capacitance of a dielectric medium (relative scale 0–99), pH was measured by using an electrochemical surface probe (range 0–14), and skin elasticity was assessed on the left shoulder by measuring the vertical deformation in a controlled vacuum applied with a circular aperture of 8 mm [4]. Cutaneous microcirculatory and capillary microscopy (using Doppler-shifted signals) were performed on capillary loops on the fingernail bed of the fourth finger with measurements taken in mm/s over a period of 15–30 min. The authors used these assessments and found, that on average, AD patients had a less acidic pH (pH 6.5) when compared to healthy individuals (pH 5.5). Furthermore, AD patients had greater skin hydration (90 in AD vs. 73 in controls), and less elasticity compared to the control subjects (48 in AD vs. 62 in controls). At baseline, the study showed that a higher number of tortuous capillaries was associated with lower mini-mental status examination (MMSE) scores in AD patients (<27 vs. 27 out of 30). In addition, these patients were treated with a cholesterinesterase inhibitor, and the outcome was measured using MMSE and clinical dementia rating sum of boxes (CDR-SB) scores [7]. The group noted that AD patients who carry the ApoE E4 allele and exhibit a larger number and higher percentage of tortuous capillaries showed better treatment outcomes at six months.

## 3. Discussion

These findings emphasize that neutral skin pH, increased hydration, and reduced elasticity, could serve as biomarkers in individuals with AD that warrants further investigation. While the study size is small as it lacks an independent validation cohort or multivariate analysis, the findings raise interesting questions. Based on the above data, we constructed a direct visual alignment of the cutaneous bio-mechano-physiological features (Figure 1). Once the differences in skin findings between AD patients and healthy controls are aligned with the relevant biological principles, we noted four paradigms.

First, we point out the shared embryologic origin of the brain and the skin. Embryologists made a groundbreaking discovery over 120 years ago regarding the connection between the ectoderm and the neuroectoderm. Hans Spemann, in 1901, introduced the term “primary neural induction” to describe the initial transformation of the ectoderm into neural tissue during neurulation [8]. This process was labeled “primary” as it was believed to be the first induction event in embryogenesis [9]. Since then, researchers have unveiled the intricate molecular mechanisms responsible for orchestrating these differentiation processes. Ectodermal differentiation towards the neural crest and neural tube pathway is influenced by fibroblast growth factor (FGF) proteins, which simultaneously inhibit bone morphogenic proteins (BMPs) [10]. On the contrary, the expression of BMP and Wnt signals prevents FGF signals in ectodermal cells, allowing them to proceed towards a non-neural ectodermal lineage, such as the epidermis [11]. Disruptions in ectodermal-neuroectodermal differentiation can lead to neural tube defects, comprising five variants: spina bifida occulta, meningocele, meningomyelocele, myeloschisis, and anencephaly. Anencephaly, for example, is characterized by an increase in alpha-fetoprotein and acetylcholinesterase levels in the maternal serum [12]. We also point out that the relationship between the brain and the skin is established in several neuroectodermal diseases including primitive neuroectodermal tumors (PNET) that arise from primitive nerve cells [13], Johnson neuroectodermal syndrome [14], and the large group of multisystem neurocutaneous syndromes (so-called phakomatoses) that affect structures primarily derived from the ectoderm such as CNS, skin, and eyes [15]. The connection between the shared ectodermal origin and the intriguing relationship between changes in higher brain function and skin properties remains uncertain at present. Nonetheless, it is important to note that a strong correlation between the development of the skin and the brain has been firmly established. This shared embryologic origin of the brain and the skin, coupled with the seemingly persistent pathophysiological relationship, holds promising potential for diagnostic approaches to be explored and utilized.

When projecting the key physiological difference back onto the corresponding histological structures, another paradigm becomes apparent. The mapping reveals that several functional systems are affected. In other words, microvascularization, innervation, eccrine excretory functions, and the extracellular matrix clearly undergo changes that have been well documented with aging [16,17,18,19,20,21]. It has previously been assumed that skin changes in AD patients might be related to aging. Specifically, in older individuals, the skin undergoes aging changes that impact microvascularization, innervation, eccrine excretory functions, and the extracellular matrix. These changes include reduced vascularity and cellularity in the dermis, decreased innervation of the skin, diminished eccrine gland function, and alterations in the composition of the extracellular matrix, including collagen, elastin, and glycosaminoglycans. These age-related changes can lead to impaired blood flow, decreased sensation, compromised sweating, and alterations in skin structure and elasticity [22]. However, the skin findings in patients with AD are distinct from age-matched controls and also seemingly incompletely characterized. In other words, these baseline pathophysiological characteristics are noteworthy because they extend the well described association between AD and several specific types of skin diseases, including bullous pemphigoid, hidradenitis suppurativa, psoriasis, skin cancer, and cutaneous amyloidosis [23].

It is important to note that the directionality of many changes follows a shared mechanical paradigm. Several studies utilizing Magnetic Resonance Elastography (MRE) [24,25,26] and shear wave dispersion ultrasound vibrometry (SDUV) [27], have demonstrated an age-related decrease in brain stiffness [28]. Even regional differences in stiffness have been reported. For example, in AD changes occur mostly in the frontal, parietal, and temporal lobes [29]. While these findings indicate that quantifying brain stiffness may serve as a valuable marker for the aging process, we point out that these alterations are seemingly mirrored in the human skin (Figure 1) [4,16,17,21,30,31,32,33]. It is important to point out that Wu et al. did not examine the brain stiffness of the patient cohort, and we are currently not aware of AD studies that directly correlated the brain findings from MRE or SDUV with bio-mechano-physiological findings in the skin; this could be a promising field of investigation for future studies.

The concrete diagnostic implications of the identified differences are noteworthy. Assessment of various functions of the human skin has been applied to several disease entities [34,35]. For example, skin testing for IgE-mediated allergic disease attempts to provoke a small, controlled, and visible allergic response [36]. Furthermore, sweat testing makes use of the fact that cystic fibrosis patients have defective sweat glands. The defective chloride channels do not allow chloride reabsorption and the concentration of chloride in sweat is therefore elevated in patients with cystic fibrosis [37]. Notably, numerous devices using microfluidic, optical, or electrochemical sensors represent novel technologies that enable wearable point-of-care applications [38]. We point out that these techniques are cost-effective, accessible, and can be easily performed repeatedly over time. Integration of skin findings with other diagnostic modalities and cognitive assessment studies seems an ideal combination of effective monitoring tests [4,21].

Shifting the focus beyond Alzheimer´s Disease, recent research emphasizes the diagnostic relevance of the brain–skin link across a broader spectrum of neurodegenerative conditions. Specifically, a recent publication by Gibbons et al. [39] features a skin test that can measure the deposition and distribution of α-synuclein, a protein involved in the neurodegenerative processes of Multiple System Atrophy (MSA) and Parkinson’s Disease (PD). Notably, autopsy studies have identified a high misdiagnosis rate [39]. Gibbons et al. discovered that phosphorylated α-synuclein was present within cutaneous autonomic nerve fibers in patients with MSA and PD; notably, these inclusions were absent in healthy age-matched controls. The extent of α-synuclein deposition was also found to be greater in MSA patients compared to those with PD. Furthermore, the topographic distribution of α-synuclein varied between patients with MSA versus those with PD. While PD patients showed a proximal-to distal-gradient, MSA patients displayed a more uniform deposition pattern. The findings by Gibbons et al. point towards the potential use of a skin biopsy to confirm a diagnosis of a synucleinopathy and distinguishing between MSA and PD. Similarly, prior studies have delineated that phosphorylated Tau Protein can be identified in the skin of AD patients [40]. Akin to the findings by Wu et al., the approach holds promise in refining diagnostic procedures, supporting research advancement, and enhancing clinical care coordination.

The discoveries mentioned indicate that there is a potential for the skin to mirror hidden brain conditions. It is conceivable that preparing cultures of skin progenitors of AD vs. healthy controls could serve as a basis for examining cellular and molecular changes otherwise present in affected brain regions [41,42,43]. Furthermore, the results emphasize a compelling brain–skin link rooted in embryology and seemingly persistent throughout life. When viewed in conjunction with the advents of modern induced pluripotent stem cell (iPSc) protocols [44,45], the above findings indicate the compelling possibility of producing iPScs from diseased vs. healthy skin and inducing neuronal differentiation to follow possible alterations in the molecular profile and functional characteristics. iPScs can be used for studying a wide range of brain diseases, allowing for disease modeling, drug screening, and potential cell regeneration, with a focus on identifying cellular phenotypes that distinguish disease-bearing cells from control cells [46]. For example, a recent study demonstrated that cultured skin cells can be stimulated with an extracellular matrix to form networks that are underdeveloped in skin cells derived from AD patients [47]. Simply put, these approaches functionally harness the pathophysiological relationship between the skin and the brain as a diagnostic and/or therapeutic tool. However, more evidence to support this speculation is needed.

## 4. Conclusions

In conclusion, we consider the findings of the relationship between the skin and the brain in AD as an exciting frontier. The shared embryologic origin highlights the interconnectedness of these seemingly distinct organs and suggests potential similarities and relationships in their development and pathophysiological properties. Therefore, an improved understanding of the relationship between the skin and the brain holds great promise for AD diagnostics. Prospective and larger scale investigations will be required to solidify the outlined brain–skin link and specify the extent to which this link can be leveraged for diagnostic applications.

## Figures and Tables

**Figure 1 ijms-24-13309-f001:**
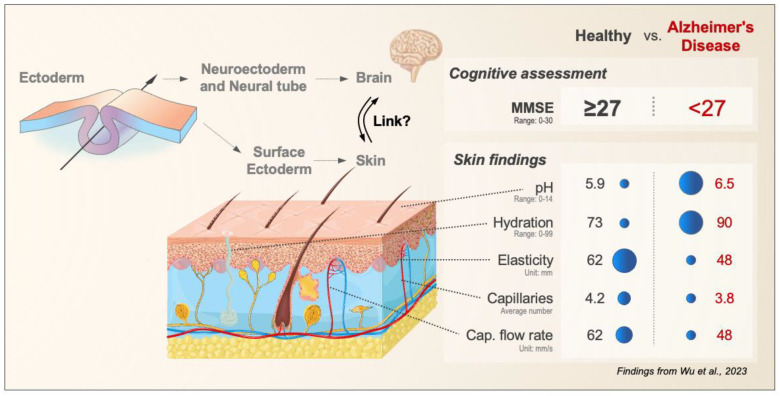
Skin–Brain Link—Potential for Alzheimer’s Disease Research. The brain and human skin share a common embryological origin as both derive from the ectodermal germ layer during early embryonic development. A recent study by Wu et al. [4] identified a series of bio-mechano-physiological differences between Alzheimer’s disease patients and healthy controls. The figure visualizes these functional alterations in relation to their corresponding histological elements. Abbreviation: MMSE, Mini-Mental State Examination; for all details of correlations, units, and ranges see text and Wu et al., 2023 [4].

## Data Availability

All data used is publicly available.

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
