# Peer review of "Exploring the Skin Brain Link: Biomarkers in the Skin with Implications for Aging Research and Alzheimer’s Disease Diagnostics"

_ijms, 2023, doi:10.3390/ijms241713309_

Round 1
Reviewer 2 Report
This is a very good commentary on recent publication: Developing Biomarkers for the Skin: Biomarkers for the Diagnosis and Prediction of Treatment 193 Outcomes of Alzheimer's Disease. Int J Mol Sci 2023 May 9;24 10. I have one advice. Please add information regarding what areas of skin were tested in the published paper to show the skin brain link for AD or Ageing.
Reviewer 3 Report
This is a commentary to a paper published early this year in another MDPI Journal. There are major issues in the manuscript. First, the construction of the commentary is difficult to follow. The authors mention both in the abstract as well as in the introduction that they comment on bio-mechanical-physiological parameters. This is not clear at all what is meant. That such parameters are involved in Alzheimer’s disease is clear since its first description by the man whose name is associated with the disease. Indeed, he described the plaques in a man with „Altersblödsinn“ (which could be equated with “Age nonsense”). So the argument of the commentary starts on the wrong foot. Then one wonders how the authors plan to assess the tissue involved in AD, which is the brain, with measuring mechanical aspects. This is also seen in the undefined use of brain stiffness. Brain stiffness can indeed be measured by MR elastography, and this is indeed mentioned later in the comment, but the original paper which is commented here did not use that technique. So it is much too earlier to correlate the skin mechanical parameters with the brain stiffness. The original paper did not do that.
However, the major problem is the figure. The graduation of the MMSE is a misrepresentation of the data from the paper. The 27 was the cut off for affected and controls in the original paper, however, the way it is presented here suggest a correlation between the MMSE results and the mechanical tests used. Only the percentage of tortuous
capillaries was correlated with the MMSE, but was not significantly different between AD and controls. Further, this was not the case with the other cognitive test used. So the values in the figure do not correspond to the actual MMSE correlation in the paper.
There is a strong need to improve English style. Just to give view examples, the very first sentence in the abstract: Diagnosing….diagnostic modalities. The second sentence is not clear bio-mechanical-physiological parameters.. of what is meant? The same in the introduction and also at several places in the commentary.
Round 2
Reviewer 3 Report
Some clarifications have been offered, but my major concern about the figure has not been addressed. The MMSE 27 as the cut off for affected and controls in the original paper is now clearly stated as the dichotomic separation in the two groups. However, suggestion of a correlation or a negative correlation between the other variables is an overinterpretation of the original data suggesting a set of continuous deta which was not the case. Suggested improvement would be to just mention increase-unchanged-decreased.
A small comment: the URL for one reference should be changed to the actuall reference number.
